# Scaling and Complexity of Stress Fluctuations Associated with Smooth and Jerky Flow in FeCoNiTiAl High-Entropy Alloy

Mikhail Lebyodkin [1], Jamieson Brechtl [2,*], Tatiana Lebedkina [1], Kangkang Wen [3], Peter K. Liaw [4] and Tongde Shen [3]

[1] Laboratoire d'Etude des Microstructures et de Mécanique des Matériaux (LEM3), Université de Lorraine, CNRS, Arts & Métiers ParisTech, 7 rue Félix Savart, 57070 Metz, France; mikhail.lebedkin@univ-lorraine.fr (M.L.); tleb1959@gmail.com (T.L.)

[2] Buildings and Transportation Science Division, Oak Ridge National Laboratory, Oak Ridge, TN 37830, USA

[3] Clean Nano Energy Center, State Key Laboratory of Metastable Materials Science and Technology, Yanshan University, Qinhuangdao 066004, China; wenkangk@163.com (K.W.); tdshen@ysu.edu.cn (T.S.)

[4] The Department of Materials Science and Engineering, The University of Tennessee, Knoxville, TN 37996, USA; pliaw@utk.edu

* Correspondence: brechtljm@ornl.gov

**Abstract:** Recent observations of jerky flow in high-entropy alloys (HEA) revealed a high role of self-organization of dislocations in their plasticity. The present work reports the first results of the investigation of stress fluctuations during plastic deformation of an FeCoNiTiAl alloy, examined in a wide temperature range covering both smooth and jerky flow. These fluctuations, which accompany the overall deformation behavior representing an essentially slower stress evolution controlled by the work hardening, were processed using complementary approaches comprising Fourier spectral analysis, refined composite multiscale entropy, and multifractal formalisms. The joint analysis at distinct scales testified that even a macroscopically smooth plastic flow is accompanied by nonrandom fluctuations, disclosing the self-organized dynamics of dislocations. Qualitative changes in such a fine-scale "noise" were found with varying temperature. The observed diversity is significant for understanding the relationships between different scales of plasticity of HEAs and crystal materials in general.

**Keywords:** high-entropy alloys; serrated flow; mesoscopic-scale plasticity; complexity; statistical analysis

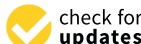



## 1. Introduction

For almost two decades, high-entropy alloys (HEAs) [1–4] have captured the attention of the materials science community as one of the possible avenues for developing materials with desirable properties, such as corrosion resistance [5–7], good fatigue resistance [8,9], irradiation resistance [10,11], excellent wear resistance [12], and high strength [13,14]. While undergoing an applied stress, HEAs often exhibit serrated, or jerky flow, which is characterized by pronounced fluctuations in the stress–strain graph [15]. Such dynamical behavior has been observed in a variety of alloy systems, including Al alloys [16,17], steels [18], bulk metallic glasses [19,20], medium-entropy alloys [21,22], and HEAs [23]. This macroscopic instability of plastic flow has been the object of numerous investigations. On the one hand, the aim of these efforts is to provide a better understanding of the causes and mechanisms of the instability in order to avoid it, as the unstable flow may be detrimental for mechanical properties [24]. On the other hand, jerky flow presents a striking example of the general phenomenon of self-organization in complex systems [25]. On the whole, the unstable flow may be due to different mechanisms. In alloys, it is typically caused by the pinning of dislocations by diffusing solute atoms (dynamical strain ageing)—the well-known Portevin–Le Chatelier (PLC) effect [26,27]. It is now generally accepted that

serrated flow can display complex behavior associated with scale invariance, a signature of a self-organized nature of the underlying deformation processes [26,27]. This complexity suggested that the approximation of stochastic dynamics of individual dislocations does not lead to a comprehensive understanding of plasticity, and, therefore, the self-organization of dislocations must also be considered. Moreover, the use of experimental methods which provide higher temporal or/and spatial resolutions, e.g., measurements of acoustic emission or evolution of local strains, proved that the macroscopically stable plastic flow also involves collective dislocation processes [28–34]. It is natural to suggest that different scale ranges assessed by particular techniques represent various aspects of a unique problem of the self-organization of dislocations. However, the corresponding investigations mostly developed separately except for several works on AlMg alloys [35–38]. These results revealed that the elementary components of both jerky and stable plastic flow consist of dislocation avalanches, as followed from the power-law statistics of deformation events.

Due to their peculiar microstructures that result in strong distortions of the crystal lattice, HEAs are particularly attractive for investigations into the self-organization of plasticity. Such studies were mostly devoted to macroscopic serrations so far [15]. The first attempt to characterize smooth plastic flow revealed power-law statistics for acoustic emission, but also unknown behaviors on a coarser scale pertaining to the local strain-rate field [39]. One important hypothesis stemming from these findings is that the apparent complexity may depend on the scale of observation. The examination of various material responses to plastic deformation is thus of great interest.

The main purpose of the present paper was to study the possible complexity associated with small stress fluctuations accompanying plastic flow, typically about two orders of magnitude smaller than the serrations caused by the PLC effect. While these fluctuations are usually ignored in mechanical tests, tacitly being attributed to noise, the detection of an inherent complexity would testify that they may reflect meaningful physical processes, and their study would therefore fill a gap between different scales of plasticity. Moreover, a challenge was to investigate whether the possible correlations within this ubiquitous "deformation noise" possess information about the macroscopic instability that may occur after some amount of deformation. This approach aiming at relating various scales of intensity of deformation processes may therefore provide valuable information on the underlying physics, not only in view of the mechanical behavior of HEAs, but more generally for the plasticity of materials.

## 2. Materials and Methods

### 2.1. Experimental Section

The $(FeCoNi)_{86}$-$Al_7$-$Ti_7$ (atomic percent) HEA samples were fabricated by arc-melting pure elements under a Ti-gettered high-purity argon atmosphere, using the same process as in Ref. [40] (see also [41] for detail on similar fabrication processes). The elements used for fabrication had a purity of at least 99.99 weight percent. All the alloy ingots were first repeatedly melted at least six times to ensure chemical homogeneity and then drop-cast into a 60 mm × 20 mm × 5 mm copper mold. The ingots were homogenized at 1150 °C for 2 h, water-quenched to room temperature ($T$), and cold rolled with a total reduction of 70% at room temperature. The sheets were then recrystallized at 1150 °C for ~1 min and furnace-cooled to 700 °C. The samples were then aged at 780 °C for 4 h and cooled to ambient temperature by quenching in water. The heat treatment occurred under vacuum (less than 0.001 MPa) using a rate of 20 °C min$^{-1}$.

For microstructure observation, the samples were mechanically ground using SiC sandpaper with a grit number ranging from 400 to 5000 and then polished with a diamond paste to 0.5 µm. The polished samples were chemically etched for 20 s in a mixed solution of aqua regia and alcohol with a volume ratio of 1:2. The microstructure was examined with the aid of optical and scanning microscopy. The analysis showed that the samples consisted of a fully recrystallized microstructure with a uniform distribution of equiaxed grains with an average size of 85 ± 32 µm, as illustrated in Figure 1a,b.

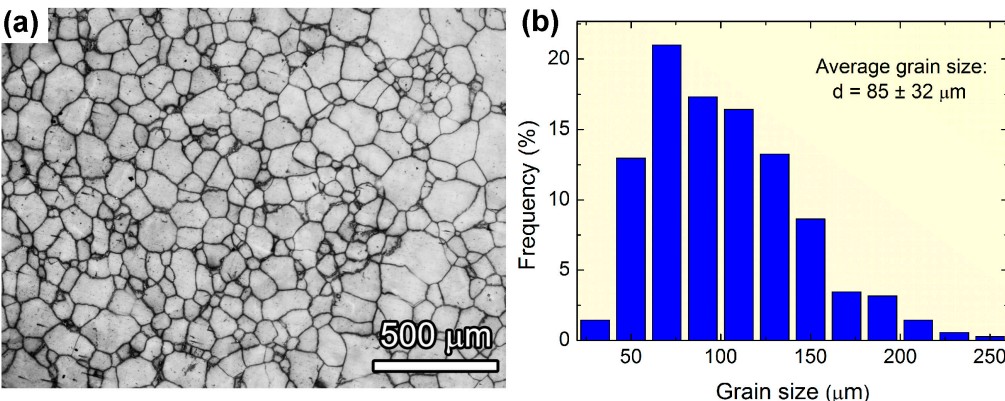

**Figure 1.** Example of (**a**) optical imagery of the microstructure of the investigated alloy and (**b**) the corresponding grain size distribution.

Flat dog-bone-shaped tensile samples with a gauge length of 5 mm and a cross-section area of 2.0 mm × 1.5 mm were cut by electrical discharge machining and polished with 2000-grit SiC papers. A computer-controlled WDW-50S MTS testing machine with a force resolution of 1 *N* was employed to investigate the tensile properties in a temperature range from 200 °C to 700 °C at a nominal strain rate of $10^{-2}$ s$^{-1}$. Two samples were tested at each temperature. Specimens were first heated to the desired testing temperature at a rate of 40 °C min$^{-1}$ and then held at the testing temperature for 10 min before tensile tests commenced. The tensile-loading direction was parallel to the rolling direction.

### 2.2. Analysis of Stress Fluctuations

To quantify the diversity and compare various kinds of small-scale stress variations, deformation curves were processed using several statistically based methods, including Fourier spectrum, Refined Composite Multiscale Entropy (RCMSE) [42,43], and Multifractal (MF) [44] formalisms. The last two approaches are described in detail in Appendices A.1 and A.2, respectively. A succinct description of their physical meaning, presented in the paragraph below, aims at providing a basis for a qualitative understanding of the presented results.

The RCMSE method is now widely used to identify the temporal complexity of signals of various nature. It investigates how the entropy of a complex time series, below also denoted as sample entropy, is changed when the signal is gradually coarsened through averaging data points over intervals with an increasing length. The coarsening is defined by a scale factor, $\tau$, which denotes the number of data points to be averaged over. The dependence of sample entropy on $\tau$ happens to be quite sensitive to the characteristic time scales which reflect correlations within the underlying physical processes. The MF formalism allows for a description of discontinuous time series possessing the feature of scale invariance, surprisingly frequent for natural objects. This characterization is performed by calculating the fractal dimension, $f$, of scale-invariant data subsets defined by a similar singularity strength, $\alpha$, which depicts the local behavior of the signal and indicates a local discontinuity when $\alpha < 1$ [44,45]. Herewith, the smaller the value of $\alpha$, the more singular is the local discontinuity.

The RCMSE and MF approaches provide complementary information. While the MF analysis is local in the sense of characterizing the clustering of local singularities, the entropy approach is global, as it blends all data without considering the instants at which the fluctuations appear. Combining both methods can therefore provide a more in-depth understanding of the dynamical behavior of plastic flow.

## 3. Results

### 3.1. Plastic Deformation Behavior

Examples of tensile stress–time curves, $\sigma(t)$, are presented in Figure 2 for four temperature values representing distinct deformation behaviors. It is seen that the temperature variation allows for investigating various cases covering both the smooth (200 °C, 600 °C, and 700 °C) and distinct types of jerky flow—the so-called type-*A* serrations at 300 °C and 400 °C and type-*C* serrations at 500 °C. Such an existence of a bounded temperature domain of unstable behavior is typical of the PLC instability, as follows from the competition of a softening effect due to an increasing dislocation mobility, and a hardening effect due to faster diffusion of solute atoms to dislocations with increasing temperature [27]. The signature of type-*A* serrations stems from the repetitive stress rises followed by relatively abrupt returns to the nominal stress level corresponding to the contour connecting the smooth portions of the deformation curve [see Figure 2b] [17,26]. In contrast, type-*C* serrations occur below the nominal level and consist of abrupt stress drops followed by slower reloading. One of the advantages of the material used for this study is a very high critical strain for the onset of type-*C* serrations, which appeared just before necking, thus allowing for a detailed analysis of the preceding smooth plastic flow. Magnification of the deformation curves in Figure 2b demonstrates that type-*A* serrations represent a different scenario when the instability develops progressively from very small amplitudes (cf. [46]).

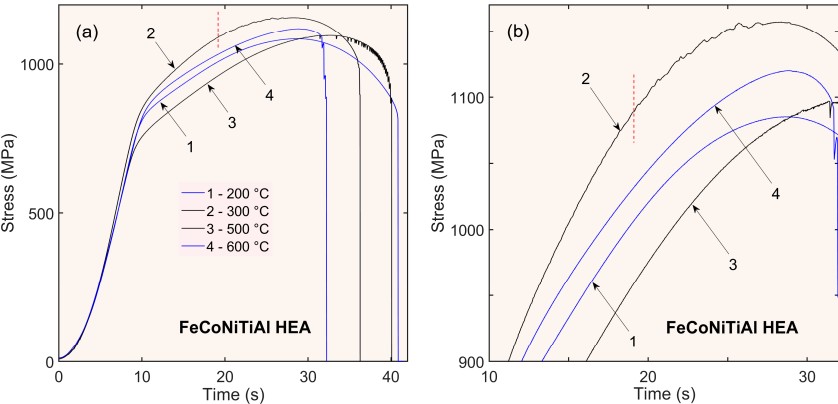

**Figure 2.** Representative examples of stress–time curves for four temperature values (200–600 °C) (**a**) and their magnification (**b**). The curves were recorded using a sampling time of 10 ms. The vertical dashed lines crossing the deformation curve recorded at 300 °C indicate the conventional way of detection of the critical strain for the onset of the PLC effect according to the first visible type-*A* serrations.

These examples also indicate a non-monotonous temperature effect on the overall deformation curves, i.e., on the material strength and ductility. This effect is likely due to the temperature dependence of the precipitation hardening caused by $L2_1$ phase precipitates (cf., e.g., [41]). However, investigation of the strengthening mechanisms which govern this slow time-scale deformation behavior goes beyond the scope of the present paper and will be discussed elsewhere.

Figure 3a–c unveil details of stress fluctuations on the deformation curves for 200 °C, 300 °C, and 500 °C, respectively, which were revealed by removing the systematic stress increase caused by the material work hardening [39]. Such detrending of stress–time curves is also necessary for the application of the methods of analysis described in Section 2.2. As a general rule, it is performed in order to avoid biasing the results of the study of the higher-frequency fluctuations by the slow evolution of stress. The detrending was implemented by subtracting a polynomial function (typically of order 6) determined in the time interval corresponding to a globally uniform plastic flow between the elastoplastic transition and the onset of necking. To visualize small fluctuations at 500 °C, the detrending was performed in the interval ending just before the first type-*C* serration. It is seen that

the fluctuations observed at 500 °C are initially irregular but increase in amplitude and acquire an oscillatory character shortly before approaching the onset of the PLC effect. The initial irregular fluctuations were visually similar to those found on the globally smooth deformation curves for 200 °C, 600 °C, and 700 °C [cf. Figure 3a]. Accordingly, the first intention was to attribute such fluctuations to valueless noise. However, a closer inspection interferes with this conjecture and imposes a quantitative analysis that constitutes the essence of this study. First, the intensity of the irregular fluctuations varies with temperature [cf. examples in Figure 3a,c], as well as between samples deformed in the same conditions. Furthermore, such a tendency appeared to increase with temperature. These observations bear witness to a nonrandom contribution to the fluctuations, which may reflect collective deformation processes. Furthermore, the example of Figure 3c illustrates a superposition of the "deformation noise" and the noise recorded during the initial idle motion of the tensile machine at 500 °C. It can be recognized that the fluctuations accompanying plastic deformation are higher in amplitude and more complex in shape than the idle noise. This comparison is deepened by the Fourier analysis illustrated in Figure 3d. Indeed, while the power spectrum density of the idle noise is concentrated above approximately 15 Hz, the spectrum of the deformation signal contains multiple intense components below this frequency. Moreover, as noted above, the fluctuations turn afterwards into higher-amplitude oscillations. The occurrence of such a "precursor" of catastrophic deformation agrees with reports on the observation of an increase in the acoustic activity before stress serrations in conventional alloys (e.g., [38,47]).

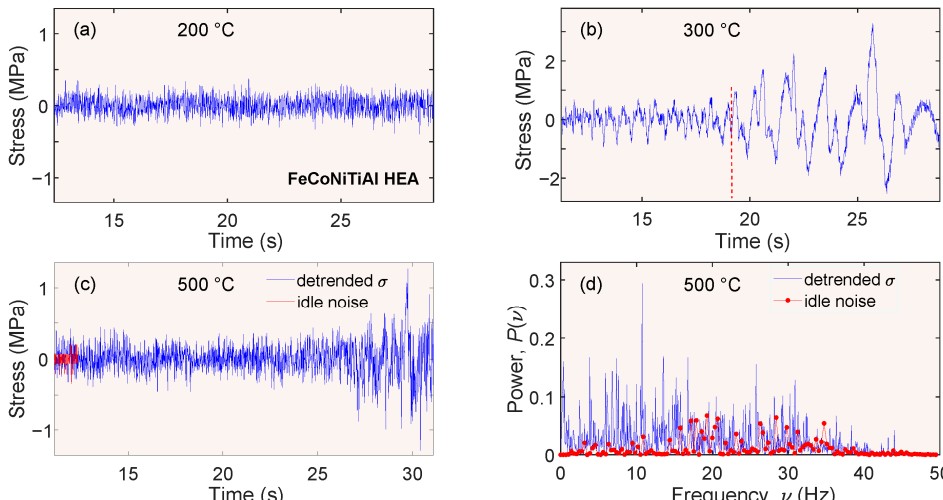

**Figure 3.** Representative examples of detrended deformation curves. (**a**) Fluctuations about a smooth deformation curve at 200 °C; (**b**) Type-*A* serrations at 300 °C. The vertical dashed line corresponds to its counterparts in Figure 2a,b; (**c**) A portion of the detrended deformation curve before the first type-*C* serration at 500 °C. Also shown is the as-recorded idle noise during a short initial time interval before the load starts increasing; (**d**) Frequency dependence of the power spectral density for the signal from Figure 3c. The spectrum was calculated over the interval $t < 27$ s, which does not include the ultimate oscillations.

The entirety of these considerations corroborates the conjecture of nonrandomness within the fine fluctuations around stable deformation curves. To complete the picture, such fluctuations faintly show up at 300 °C and 400 °C. Here, they are overshadowed by oscillations occurring almost immediately upon the elastoplastic transition and finally turning to macroscopic type-*A* serrations [Figure 3b].

### 3.2. RCMSE and MF Analysis

The above conjecture was further corroborated by virtue of the quantitative approaches provided by the RCMSE and MF methods. Figure 4 displays examples of the RCMSE($\tau$)

dependences for a set of specimens cut from the same sheet sample. The entropy was cal­culated over the entire detrended portions of the deformation curves, except for $T = 500\,°C$. To compare the results obtained for this temperature to other cases of smooth flow, the analyzed interval was limited to the initial irregular fluctuations, similar to the above example of Fourier analysis [$t \leq 27$ s in Figure 3c]. This interval is further referred to as region I for $T = 500\,°C$.

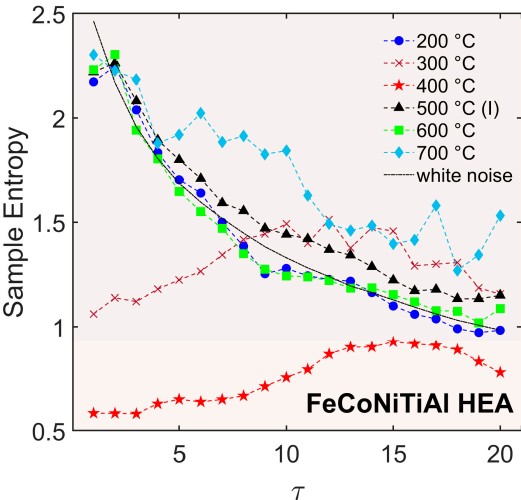

**Figure 4.** Sample entropy dependences on the scale factor, $\tau$, for detrended deformation curves recorded at different temperatures. The mark (I) refers to the analysis of the initial part, $t \leq 27$ s, of the deformation curve at 500 °C.

It can be seen that the smooth flow corresponds to generally descending RCMSE($\tau$) dependences. Therewith, the curves found at both 200 °C and 600 °C are close to that for white noise. However, as the deviations from the latter occur at the same $\tau$ values for both temperatures, in particular in the middle range, they may indicate the presence of (weak) correlations. Moreover, the dependences obtained for 500 °C and 700 °C discernibly deviate from random behavior at all scales. As the relative increase in the sample entropy reflects an increasing complexity (see examples in Appendix A.1), the slower downward trend of these curves with regard to the case of white noise, especially at 700 °C, indicates stress fluctuations with a greater degree of complexity. Moreover, the greater sample entropy values for larger $\tau$ bear witness that the fluctuations at these temperatures retained greater information content when averaged over more points, further showing that they were more dynamically complex as compared to the white noise. These results quantitatively corroborate the conjecture of collective deformation processes taking place on a mesoscopic scale during smooth plastic flow (cf. [39]).

The examples of RCMSE($\tau$) dependences obtained for the conditions of type-*A* behav­ior of the PLC effect show a trend qualitatively different from that for the stress fluctuations about macroscopically smooth deformation curves. Therewith, although the curves for 300 °C and 400 °C are offset from each other, which indicates quantitative variations be­tween stress serrations of the same type (the possible reason for this gap will be further clarified with the aid of the MF analysis), the qualitative trend is similar for both temper­atures. Namely, the sample entropy curves grow towards the middle $\tau$ range. Such an ascending drift indicates the dominance of the complexity associated with type-*A* stress serrations, which is disclosed when the higher-frequency fluctuations are smoothened out due to scale coarsening.

Figure 5 exhibits examples of "singularity spectra", $f(\alpha)$, for distinct deformation behaviors. To avoid superposition of closely passing dependences and provide readability of the plot, the singularity spectra are not shown for 600 °C and 700 °C. Their relative positions are clear from the data presented in Figure 6. For comparison with the MF features

of the deformation curves, an apparent $f(\alpha)$ dependence is also traced for a generated white noise. As described in Appendix A.2, to assure an equitable comparison with $f(\alpha)$ of white noise, which theoretically must collapse to a single point with nonfractal dimension $f = 1$, an apparent $f(\alpha)$ curve was calculated for a random signal generated over a time interval of the same length as the typical deformation curve. It can be seen that although the spectrum does not collapse, it is much narrower than the curves found for the experimental time series. A similar check was also made for the noise recorded in the idle deformation state. In this case, no reliable scaling allowing for determination of the MF spectrum was found. This absence of a trivial scaling for a presumably uniform signal is most likely caused by accidental outliers present in the idle noise, which may break the possible scaling.

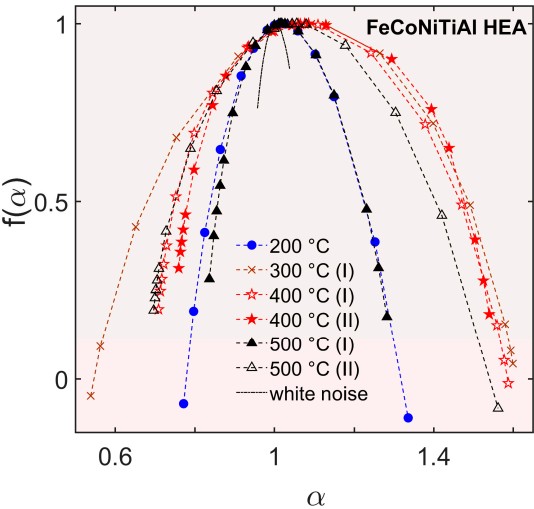

**Figure 5.** "Singularity spectra", $f(\alpha)$, of the deformation noise representing qualitatively different behaviors. The Roman numerals after the $T$ values indicate the cases when the calculation was performed separately for the initial (I) and later (II) portions of the signals.

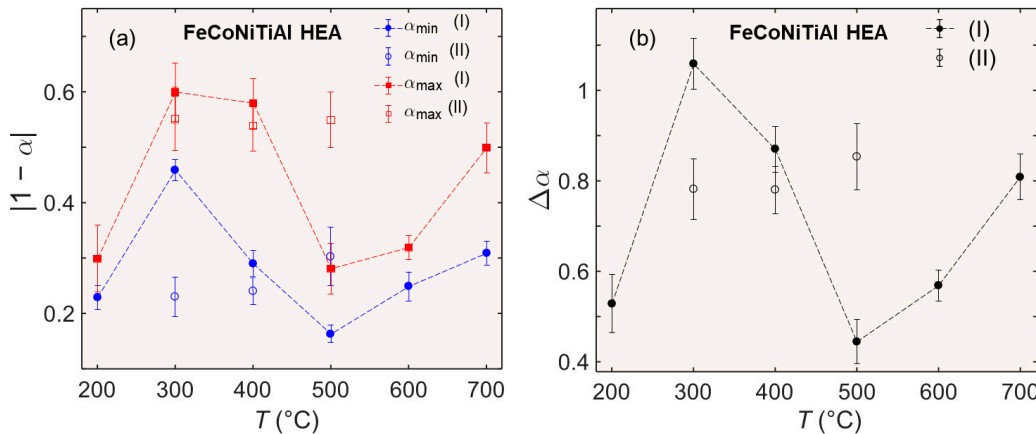

**Figure 6.** (**a**) Position of the edges of the singularity spectra, $\alpha_{\min}$ and $\alpha_{\max}$, relative to unity. (**b**) The corresponding width of the spectrum at the level of $f(\alpha) = 0$. The Roman numerals have the same meaning as in Figure 3.

The dependences presented in Figure 5 illustrate that reliable MF behavior was found for stress fluctuations at all temperatures. Thus, the local features of the analyzed signals also confirm the presence of an intrinsically nonrandom deformation noise. The exact shape of the spectra should be taken with caution because the interval of scaling was relatively short and varied between 1 and 1.5 orders of magnitude of $\delta t$. The shortest scaling length was found for the low-amplitude irregular stress fluctuations. Consequently, the example

of an MF dependence at 200 °C shows certain flaws manifested, e.g., in the unphysical negative values of $f(\alpha)$ at the edges of the dependence. A similar drawback is seen for the left wing of the MF spectrum denoted as 300 °C (I) in Figure 5, which was calculated for the time interval before the onset of the macroscopic type-*A* serrations, i.e., to the left of the vertical dashed line in Figures 2 and 3c. In contrast, the type-*A* serrations, as well as the fluctuations observed before the onset of type *C* instability at 500 °C, allowed for a reliable determination of MF spectra, at least as far as the most statistically significant left branches (see Appendix A.2) are concerned. More exactly, the error in determination of $\alpha_{\min}$ did not exceed 0.06, and the error in $f(\alpha_{\min})$ was less than 0.12 in all such cases. It is expected that further increases in the time and force resolution will allow for improving the determination of the MF behavior of low-amplitude signals.

Despite these quantitative uncertainties in some of the calculated MF spectra, the changes in $\alpha$ induced by the temperature variation, and summarized in Figure 6, qualitatively agree with those in the sample entropy (Figure 4) and allow for a consistent explanation. In this concern, specific attention can be drawn to the width of the MF spectra, $\Delta\alpha = \alpha_{\max} - \alpha_{\min}$ [Figure 6b], which characterizes the signal heterogeneity regarding its local singularity (cf. [17]): a wide spectrum corresponds to a high inhomogeneity, while $\Delta\alpha$ would be equal to 0 for the ideal case of a uniform fractal. In the cases when $f(\alpha)$ descends below zero (see Figure 5 and the comments in the previous paragraph), the extreme $\alpha$ values were determined at the level of $f = 0$.

It is seen that the width of the MF spectra is narrow at 200 °C and 600 °C, i.e., in the cases for which the entropy analysis (see Figure 4) has revealed a close-to-random behavior. In addition, the initial irregular fluctuations at 500 °C (I) also show a low heterogeneity although the corresponding behavior is not random, as certified by the RCMSE analysis. Significantly, $f(\alpha_{\min})$ takes on a noticeably positive value for these fluctuations (see Figure 5), which indicates a high degree of uniformity associated with the most singular events [26], which may be attributed to a tendency to a periodic occurrence of such discontinuities. The spectrum width becomes quite high for the part 500 °C (II) preceding the onset of type-*C* serrations. It can also be seen in Figures 5 and 6a that $\alpha_{\min}$ decreases considerably in comparison with its value in region 500 °C (I), which testifies to an increase in the maximum local singularity of the fluctuations. Therewith, $f(\alpha_{\min})$ is also positive in region (II), in consistence with the occurrence of large stress fluctuations directly visible on the detrended signal in Figure 3c. Finally, the deformation noise observed at 700 °C is also characterized by a high heterogeneity and local singularity, as reflected in a relatively large $\Delta\alpha$ and small $\alpha_{\min}$, and could also be expected from the RCMSE analysis. Overall, the increase in the intensity of stress fluctuations, be it due to a temperature increase (700 °C), or due to changes emerging after some deformation [500 °C (II)], led to stronger heterogeneity and singularity in terms of MF analysis. As far as $f(\alpha_{\min})$ is concerned, it is zero at 700 °C, which is similar to what was observed at the other two temperatures outside the PLC domain.

The initial behaviors preceding the onset of type *A* serrations, denoted in Figure 5 as 300 °C (I) and 400 °C (I), manifested a high heterogeneity, which may be explained by the superposition of the deformation noise present at all temperatures with the oscillatory modes occurring in type-*A* conditions. The most singular (the least $\alpha_{\min}$) and heterogeneous (the greatest $\Delta\alpha$) behavior was found at 300 °C, which showed oscillations with relatively low amplitudes, so that the deformation noise could have a notable influence on the MF scaling behavior [see Figure 3b]. The oscillations were more intense at 400 °C such that the domination of one kind of signal may explain a narrower spectrum than at 300 °C. This difference between the signals at 300 °C and 400 °C may also be responsible for the overall lower sample entropy curve at 400 °C, as discussed above (Figure 4). In contrast to these macroscopically smooth regions I, the large oscillations corresponding to type-*A* serrations (regions II) rendered very similar $f(\alpha)$ curves for both temperatures (cf. $\alpha$ and $\Delta\alpha$ values in Figure 6). Therewith, it can be seen that the heterogeneity and the maximum local singularity diminish upon the onset of type-*A* serrations, in agreement with their regular character, usually ascribed to a repetitive propagation of deformation bands along

the specimen [17,26,27]. Also, the curve traced in Figure 4 for 400 °C (II) illustrates an increasing $f(\alpha_{\min})$, as could have been expected from the discussion of the above examples.

## 4. Discussion

The conventional approaches to the analysis of deformation curves aim at characterization of their average behavior and overlook tiny stress fluctuations, tacitly attributing those to experimental noise. The main objective of the present paper was to verify whether this "noise" can carry information on real deformation processes at a scale intermediate between the macroscopic plastic flow and the scales provided by experimental methods with a drastically higher resolution. The presented results testify that although random noise is important in the small-scale stress fluctuations measured using standard tensile machines without special precautions, they also contain "true" events which can be unveiled due to adapted mathematical methods. It can also be expected that further progress in the understanding of the deformation processes controlling this scale can be obtained by searching for ways to reduce the instrumental noise and optimize the time and force resolution of the recording system.

The results obtained allow for several conjectures on the complexity of the underlying collective deformation processes, which can help in defining the further development of such studies. Let us first recall that due to the combination of different approaches to complex signals, correlation of deformation processes was detected for all deformation conditions. However, these manifestations were weak at 200 °C and 600 °C. This result is not surprising because these temperatures are situated outside the range where macroscopic instabilities—an irrefutable signature of self-organization—are observed. Furthermore, the closeness of both the entropy curves and MF spectra for 200 °C and 600 °C suggests that similar mechanisms underlie the dynamics of plastic flow throughout this temperature interval. Nevertheless, the entropy analysis indicated a higher complexity of the visually similar stress fluctuations at the intermediate $T$ = 500 °C (Figure 4). As this temperature corresponds to the domain of the PLC instability, this discrepancy implies that the conditions of the dislocation–solute interaction leading to the macroscopic instability may also affect the fine-scale collective processes before the onset of the instability [38,48]. This conjecture is corroborated by the results of MF analysis at 500 °C, which showed a burst in the fluctuation heterogeneity and local singularity soon before the first type-*C* stress drop (Figure 6).

This suggestion is also consistent with the observation of a tendency to positive values of $f(\alpha_{\min})$ in both regions analyzed at 500 °C (Figure 5), which attracts attention to an additional aspect. Indeed, positive $f(\alpha_{\min})$-values were also observed at all temperatures within the PLC domain, including 300 °C and 400 °C corresponding to type *A* behavior. Therefore, this tendency may be intrinsic for the PLC effect that implies a recurrent occurrence of macroscopic stress serrations representing the regions with the strongest discontinuities on the deformation curve [49]. On the contrary, the observation of a comparable trend for the visually irregular stress fluctuations at 500 °C (I) is unexpected because such a trend does not show up for similar "deformation noises" at all temperatures beyond the PLC domain. Therefore, this finding allows for a supposition that the conditions leading to the PLC effect may also promote a tendency to recurring behavior in the fine-scale deformation processes. Vice versa, this peculiarity is a candidate for further investigation as a possible indicator of approaching a macroscopic instability.

As far as type-*A* serrations are concerned, both the RCMSE and MF approaches unveiled behaviors qualitatively different from those found in the absence of the PLC effect. The results of the analysis prove that at 300 °C and 400 °C the complexity is dominated by the macroscopic serrations. In this connection, it is noteworthy that according to the data for 400 °C, at which it was suggested it would be less affected by the deformation noise than its counterpart at 300 °C, both the maximum local singularity and the overall heterogeneity displayed relatively small differences in the regions (I) and (II). This similarity allows for a hypothesis that the stress oscillations preceding the type-*A* instability are essentially caused

by the same deformation mechanism as the macroscopic instability itself. This assumption has a particular meaning for the understanding of the development and complexity of macroscopic instabilities (cf. [46]) and will require a further verification.

Finally, a sharp increase in the complexity occurred at 700 °C, as stemmed from both the RCMSE and MF results (Figures 4–6), although no PLC instability was observed at this temperature. Such nonmonotonous changes with varying temperature in the investigated range were also corroborated by the analysis of complementary cumulative distribution functions [50], although a detailed comparison of the histograms of statistical distributions would require more experiments and will be presented elsewhere. The totality of these data bears witness that, independently of the PLC instability, the temperature variation may itself affect the collective deformation processes on a mesoscopic scale, be it due to a direct effect on the dislocation dynamics or to microstructure transformations leading to different conditions for the dislocation motion. This suggestion implies the necessity of detailed microstructural analyses in order to interpret the underlying physical mechanisms. It also suggests that one of the possible ways for a further verification of the mechanism of the temperature effect on the fine-scale plasticity will be studying materials with distinct microstructures. Therewith, the behavior of HEAs may essentially differ from that of conventional materials because of the presence of particular sources of resistance to dislocation glide, such as local changes of the order of atoms of different elements [51]. Moreover, such a short-range chemical ordering may also modify the mechanism of jerky flow [52]. Accordingly, further investigations should envisage the comparison of the complexity features for HEAs with different compositions, as well as with regard to low-entropy alloys, supported by characterizations of the microstructure evolution.

## 5. Concluding Remarks

In summary, the first results of the analysis of fine-scale stress fluctuations during tensile deformation of an HEA allow an extension of the conclusion on the intrinsically collective nature of the underlying dislocation dynamics. This conclusion was earlier advanced on the basis of investigations of diverse materials using higher-resolution techniques, such as acoustic emission or visualization of the local strain fields. The analysis of stress fluctuations shows that this conjecture covers an even larger range of small scales. It turns out to be a general feature that the macroscopic plastic flow is not a result of averaging over uncorrelated movements of individual dislocations, but over collective dislocation processes taking place on a hierarchical range of mesoscopic scales. Fundamentally, the data obtained testify that the collective effects may manifest qualitatively distinct dynamics in different scale ranges.

**Author Contributions:** T.S. conceived and designed the experiments; K.W. fabricated samples and performed the experimental work; M.L., J.B. and P.K.L. conceived the theoretical approach; M.L., J.B. and T.L. conducted the analyses of the experimental data; M.L., J.B. and P.K.L. helped write the manuscript and carried out text editing. All authors have read and agreed to the published version of the manuscript.

**Funding:** M.L. acknowledges the support from the French State through the program "Investment in the future" operated by the National Research Agency, in the framework of the LabEx DAMAS [ANR-11-LABX-0008-01]. P.K.L. very much appreciates the support from (1) the National Science Foundation (DMR—1611180, 1809640, and 2226508) with program directors, J. Madison, J. Yang, G. Shiflet, and D. Farkas and (2) the US Army Research Office (W911NF-13–1-0438 and W911NF-19–2-0049) with program managers, M.P. Bakas, S.N. Mathaudhu, and D.M. Stepp. T.S. thanks the support from the National Natural Science Foundation of China (Grant Nos. 11935004 and 51971195).

**Data Availability Statement:** The data presented in this study are available on request from the corresponding author. The data are not publicly available due to legal or ethical reasons.

**Conflicts of Interest:** The authors declare no conflict of interest.

## Appendix A. Analytical Methods

*Appendix A.1. Refined Composite Multiscale Entropy Analysis*

The Refined Composite Multiscale Entropy (RCMSE) approach is based on the notion of entropy as a measure of the amount of information contained in a complex signal [43]. More specifically, it measures the deviation of the data points regarding a tolerance value based on the standard deviation of the data and evaluates the changes in the measure when the processed signal is gradually coarsened by averaging data points over intervals with an increasing length. It occurs that the entropy dependence on the coarsening degree (below denoted as a scale factor, $\tau$) shows qualitatively different trends for noises of different nature [42,53]. For example, the entropy of white noise monotonously decreases with coarsening in agreement with an uncorrelated nature of such noise [53,54]. The blue noise mentioned in the main text, whose specific feature is power spectral density increasing with the frequency, shows an even faster decrease in the entropy than the white noise characterized by a constant spectral density. On the contrary, Brownian (red) noise, which exhibits a decreasing power spectrum, is characterized by increasing entropy with coarsening. This trend reflects a complexity associated with the lower frequencies and manifests when the higher-frequency noise is suppressed by the coarsening. Scale-invariant behaviors show flat entropy dependences, thus reflecting the presence of complexity at all scales [55,56]. Such a strong diversity of the entropy behavior therefore qualifies the RSMSE technique as a powerful tool for the evaluation of real complex objects.

The following discussion provides a recipe for performing the RCMSE analysis on the serrated-flow data, which follows the method discussed in [43,57]. As an initial step, the stress–strain data in the strain-hardening regime are fit using a 6th order polynomial and then the underlying trend is removed [26,58]. The procedure is then applied to the detrended time series data (see the main text). From here, one constructs the coarse-grained time series using the following equation:

$$ y_{k,j}^{\tau} = \frac{1}{\tau} \sum_{i=(j-1)\tau+k}^{j\tau+k-1} x_i \quad ; \; 1 \; \leq \; j \; \leq \; \frac{N}{\tau} \quad 1 \leq k \leq \tau \tag{A1} $$

where $\tau$ is the scale factor, $k$ is an indexing factor which designates where in the series to begin the coarse-graining, $x_i$ the $i$th point of the detrended time series data, and $N$ is the total number of data points of detrended data. Here, the length of the coarse-grained time series $y_{k,j}^{\tau}$ is $N/\tau$ [59]. After constructing the coarse-grained time series, create the template vector, $y_{k,i}^{\tau,m}$, of dimension $m$:

$$ y_{k,i}^{\tau,m} = \left\{ y_{k,i}^{\tau} \; y_{k,i+1}^{\tau} \; \cdots \; y_{k,i+m-1}^{\tau} \right\} \quad ; \quad 1 \; \leq \; i \; \leq \; N-m \; ; 1 \; \leq k \; \leq \tau \tag{A2} $$

Here each $y_{k,j}^{\tau}$ term in the brackets of Equation (A2) is determined using Equation (A1). Next, determine the $k_{\text{th}}$ template vectors of dimension $m$:

$$ y_k^{\tau} = \left[ y_{k,1}^{\tau} y_{k,2}^{\tau} \cdots y_{k,i+m-1}^{\tau} \right] \tag{A3} $$

The number of matching sets of distinct template vectors for each $k$ is then evaluated using the following equation [60]:

$$ d_{jl}^{\tau,m} = \left\| y_j^{\tau,m} - y_l^{\tau,m} \right\|_{\infty} = \max \left\{ \left| y_{1,j}^{\tau} - y_{1,l}^{\tau} \right| \cdots \left| y_{i+m-1,j}^{\tau} - y_{i+m-1,l}^{\tau} \right| \right\} < r \tag{A4} $$

Here, $d_{jl}^{\tau,m}$ is the infinity norm, and $r$ is a limiting factor usually chosen as 0.15 times the standard deviation of the data [61] to make sure that the results are independent of the variance of the time series data [56,62]. Based on Equation (A4), two vectors match when the norm is less than $r$.

The process, as described above, is then performed for vectors of size $m + 1$. From here, the total number of matching vectors, $n_{k,\tau}^m$ and $n_{k,\tau}^{m+1}$, can be determined by taking the sum from $k = 1$ to $\tau$. These values are then used to evaluate the RCMSE, or sample entropy, for the detrended time series data:

$$RCMSE(\boldsymbol{X},\ \tau,\ m,\ r) = Ln\left(\frac{\sum_{k=1}^{\tau} n_{k,\tau}^m}{\sum_{k=1}^{\tau} n_{k,\tau}^{m+1}}\right) \tag{A5}$$

*Appendix A.2. Multifractal Analysis*

A rigorous mathematical description of the multifractal (MF) formalism can be found in numerous reviews and books (e.g., [26,44]). Particularly, Ref. [26] considers its application to the investigation of jerky flow. Below, only one aspect of this formalism is presented, which describes the underlying physics and the calculation of the so-called singularity spectrum used in the present paper to compare the complexity of the deformation noise for different experimental conditions.

The fractal analysis refers to the way of characterization of the geometry of porous objects, which porosity is not random but follows a construction rule giving rise to a scale invariance [44,45]. The potential of application of this concept to plasticity problems was first noticed in relation to the analysis of jerky flow [17]. Here, the jerky flow was thought to transmit the intermittent nature of the plastic instability and thus occur on a "porous" geometrical support. To understand the notion of a fractal, one may first consider the trivial case of self-similarity of a non-porous object. The dimensions of such an object obey scaling laws with integer exponents with regard to the measuring ruler division, $l$. For example, the number of the divisions covering an interval is proportional to $l^{-D}$ (the relationship is rigorous in the limit $l \rightarrow 0$), where the exponent, $D = 1$, is the topological dimension of the one-dimensional space on which the interval is defined. For a scale-invariant porous object, the number of the filled sites can be characterized by a similar expression where the exponent is no longer an integer and is instead referred to as the fractal dimension, $f$. Such a simple approach is, however, insufficient for heterogeneous objects. Besides the possible multiplicity of construction rules for the geometrical support itself, real objects carry nonconstant physical quantities (e.g., mechanical stress in the case of plastic deformation [63]) so that complexity may arise even for a quantity defined on a continuous support. A direct description of such complex objects would be unintelligible, needing a large table of numbers (infinite in the mathematical limit) providing the positions of all non-empty sites and the corresponding values of the physical quantity. However, it occurs that many natural phenomena possess the property of scale invariance in a statistical sense [44]. This feature made it possible to extend the fractal approach to the MF formalism [64]. The latter is based on the examination of the scaling of a local probabilistic measure, $\mu$, which is defined to characterize the intensity of variation of the signal. For detrended time series, like those presented in Figure 2a–c of the paper, the idea is to cover the analyzed time interval with a grid with a step, $\delta t$, and determine the singularity strength, $\alpha$, for each location from the following relationship which describes the scaling of the local measure:

$$\mu(\delta t) \sim \delta t^{\alpha} \text{ for } \delta t \rightarrow 0, \tag{A6}$$

Next, find the data subsets corresponding to similar $\alpha$ values, and calculate the fractal dimension, $f(\alpha)$, for each of such interpenetrating fractal subsets. The meaning of $\alpha$ is that it describes how fast the density of the measure diverges in the limit $\delta t \rightarrow 0$. Indeed, as follows from Equation (A6), the density of the local measure, $\mu(\delta t)/\delta t \sim \delta t^{\alpha-1}$, indicates a local discontinuity when $\alpha < 1$. In particular, $\alpha_{\min}$ corresponds to the location with the densest measure.

This description explains the underlying physical concept but is impractical for implementing the calculations. In the present investigation, we used a direct method of

calculating $f(\alpha)$ that was suggested in [65] and stems from scaling relationships for the $q$th moments of the probability distributions, $\Sigma(\delta t, q)$, defined for a normalized measure,

$$\widetilde{\mu}_i(\delta t, q) = \mu_i^q / \sum_j \mu_j^q :$$
$$\Sigma_\alpha(\delta t, q) = \sum_i \widetilde{\mu}_i(\delta t, q) ln\mu_i(\delta t) \sim \alpha(q) ln\delta t \tag{A7}$$
$$\Sigma_f(\delta t, q) = \sum_i \widetilde{\mu}_i(\delta t, q) ln\widetilde{\mu}_i(\delta t, q) \sim f(q) ln\delta t$$

where $i$ enumerates the $\delta t$ boxes. Variation of $q$ brings attention to different subsets of the analyzed signal. Indeed, taking large positive $q$s makes the largest values of the local measure dominate in Z functions [Equation (A7)], whereas large negative $q$s mark the least values of the measure. Making $q$ vary between the extremes allows all the different subsets to be scanned.

Several practical issues should be noted. First, the choice of $\mu(\delta t)$ that would reveal the underlying scale-invariant ordering is a kind of art because various choices are usually possible [44]. In the present paper, $\mu(\delta t)$ was calculated as a sum of the absolute values of the detrended stress, $|\sigma(t_j)|$, where the index, $j$, corresponds to the data points within the considered $\delta t$ box. Furthermore, as the least measured values are the most overshadowed by undesirable noise, some authors limit the analysis to the positive $q$s. Nevertheless, Figure 5 of the paper shows that the two branches of the $f(\alpha)$ dependence, which correspond to different signs of the moment $q$, change with temperature in a similar manner. It can thus be suggested that the right descending branch corresponding to the negative $q$s is determined correctly, providing that the negative $q$ is not varied in too large of a range. In the present investigation, it was swept between 12 and $-7$. Finally, Figure A1 illustrates the feasibility of the MF analysis by comparing the scaling of the partition functions, $\Sigma_\alpha(\delta t, q)$, for one of the real detrended time series and for a random signal over a time interval of the same length as the typical deformation curve. The necessity of such a precaution is obvious from plot (b). The figure shows that although the partition functions rapidly converge to the same slope, which agrees with the nonfractal structure of the uniform noise corresponding to a singularity spectrum consisting of a single point, (1, 1), the dependences form a narrow fan with decreasing the box size towards the smallest time step. Even though this fan collapses for a long enough random signal, it may still give rise to an apparent singularity spectrum for a short-time series. Accordingly, such an apparent spectrum was calculated and compared to the spectra obtained for the deformation curves [cf. Figure 5 in the main text]. Finally, plot (a) demonstrates that the MF scaling is conveniently determined for the detrended signal.

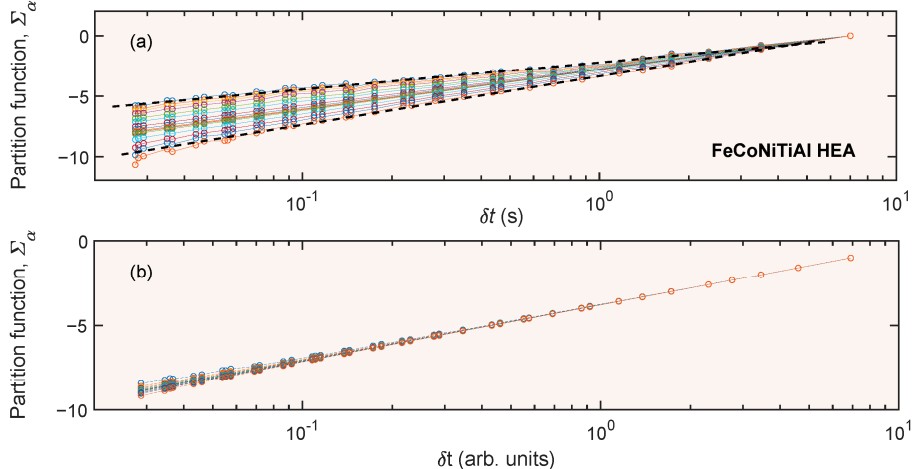

**Figure A1.** Example of a family of partition functions, whose scaling gives estimates of $\alpha$ values. Different lines correspond to different $q$ in the range from 12 to $-7$ (**a**) Results of calculations for a detrended signal at 300 °C (the initial part denoted in the paper as I). (**b**) Similar calculations for a generated random noise.

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
