# Peer review of "Scaling and Complexity of Stress Fluctuations Associated with Smooth and Jerky Flow in FeCoNiTiAl High-Entropy Alloy"

_metals, doi:10.3390/met13101770_

Round 1
Reviewer 1 Report
The manuscript presents an analysis of the fine-scale stress fluctuations observed during tensile deformation of a high-entropy alloy with particular focus on dislocation dynamics. The research report is well written and formulated. The experimental camping is complete and provides a robust support to the conclusions. In the light of the above, I suggest to accept this work for pubilcation.
Author Response
We are very grateful for a completely positive opinion on our manuscript.
Reviewer 2 Report
The present article is interesting and having new and knowledgeable information. I can recommend this article for its consideration of publications in Metals. Before, the authors have to address the following points.
1. The present abstract is not clear and it is not giving any information about method of deformation study which the authors have carried out. It needs to be revised with more details
2. The authors have mentioned about serrated behaivor of HEA in introduction part. This is related to samples or the mechanical behaviour? Need clarification. If samples exhibit serrated behaviour mean, doubt about it mechanical strength and then, its applications.
3. Some more literature related to various processing routes and selection of arc melting method of present HEA system is to be incorporated as it is missing in present form of introduction part.
4. In methodology, the use of “Ti-gettered” is not giving proper meaning. Need to be checked
5. The measured grain size from SEM images are to be incorporated in results and discussion part.
6. Further, why the authors haven’t incorporate any microstructures of their samples before testing and after?
7. In addition, the fracture surface images are also to be added
8. A schematic representation about the present research may incorporate in methodology section which may attract the readers
9. Photograph of prepared samples and used equipment’s may incorporate
10. To analyse the stress fluctuations, the authors have used RCMSE which mean a type of software or coded language. Need clarification
11. How many trials in each conditions were tested? It is not addressed
12. Based on Fig. 1, why the tensile tress curve shifted up with increasing temperature and then dropped with further increasing of temperature. What are the mechanisms behind this? It has to be elaborately discussed
13. In general, at elevated temperature, scale will form which may the reason for formation of serrated curve. It has to be addressed properly
14. In overall, the present manuscript is not discussed with the corresponding microstructures.
Language is acceptable level
Reviewer 3 Report
In the references, please add the pages related with the article. They are several without the pages definition.
In page 2, line 77 and line 79, please use a common way of writting the temperature, 1,150 ºC or 1150 ºC
Reviewer 4 Report
The manuscript presents new original results of experimental investigation of stress fluctuations during plastic deformation of (FeCoNi)86-Al7-Ti7 high-entropy alloy (HEA) at a nominal tension strain rate of 10-2 s-1 in a temperature range from 473 K up to 973 K. The several statistically-based methods of analyses such as Fourier spectrum, Refined Composite Multiscale Entropy (RCMSE), and Multifractal (MF) analyze technique were used for determination of small-scale stress variations parameters on the deformation curves under different loading conditions.
It was found that the spectrum of the deformation signal contains many intense components below the frequency of 15Hz.
It was assumed that the interaction of dislocations with precipitates in the VEA influences the formation of collective processes of small-scale self-organization that arise before macroscopic instability.
The obtained results of the analysis of experimental deformation curves indicate that macroscopic plastic flow is the result of averaging over exactly collective dislocation processes occurring in the hierarchical range of mesoscopic scales. Moreover, collective effects in different scale ranges can exhibit qualitatively different dynamics.
The manuscript is well structured, illustrated and contains a satisfactory list of references.
The conclusions are based on the analysis of the obtained results.
The results of the manuscript will be of interest to a wide range of specialists, graduate students and students who carry out research into the physical mechanisms of deformation of HEA, including the influence of the collective effects of dislocation dynamics on the mechanical behavior of HEA.

Author Response
Thank you for the high appreciation of our study.
Reviewer 5 Report
The manuscript “Scaling and complexity of stress fluctuations associated with smooth and jerky flow in a FeCoNiTiAl high-entropy alloy” addresses an actual problem related to design of high entropy alloys and study of their deformation behavior. The investigation of stress fluctuations under plastic deformation of a FeCoNiTiAl alloy in a wide temperature range revealed both smooth and jerky flow. It was shown that even a macroscopically smooth plastic flow is accompanied with nonrandom fluctuations. It is interpreted in terms self-organized dynamics of dislocations. Qualitative changes in the fine-scale “noise” were established at varying temperature. IT is stated that the obtained results are of importance for understanding the input of different scales of plasticity in ensuring HEA’s deformation behavior under loading.
The manuscript falls within the scope of the journal of Metals.
The level of English language is OK.
The state of the art is well characterized with citing enough number of the relevant papers.
The experimental procedure is clearly described; the experiments might be reproduced elsewhere.
The experimental results are clearly characterized.
Unfortunately, the discussion is given together with experimental evidences. Since the number of results is limited, their interpretation should be presented separately. This will make their understanding more evident.
The Conclusion is either a summary of conducted experiments or prospects of the study. There is neither generalization of obtained results nor their numerical characterization.
The manuscript requires major revision. The following aspects are to be addressed by the authors.
Page 2, line 42 “Moreover, the use of higher-resolution methods, e.g., measurements of acoustic emission…”. Acoustic emission is an efficient NDT technique quite sensitive even to collective motion of dislocations; however, it does not possess high spatial resolution.
Page 4, line 132. “The signature of type-A serrations stems from the repetitive stress rises followed by relatively abrupt returns to the master curve”. The term master curve is not introduced in the paper.
In addition, the effect on testing temperature on general deformation response of the alloy is not described. However, the temperature affect both mechanical properties and serrations of both scales.
Page 6, lines 207-220 “Several aspects should be noted. First, the closeness of the entropy curves for 200 °C and 600 °C suggests that similar mechanisms underlie the dynamics of dislocations throughout this temperature interval. Nevertheless, as noticed above, a deviation from this behavior is detected at the intermediate T = 500 °C. As this temperature corresponds to the domain of the PLC instability, this deviation implies that the conditions of the dislocation-solute interaction leading to the macroscopic instability may also affect the fine-scale collective processes before the onset of the instability [48,49]. On the other hand, nonrandom behavior is also found for T = 700 °C, which corresponds to stable plastic flow. Therefore, in itself, the change in the dislocation mobility with temperature may affect their collective motion, even if the temperature effect is not monotonous, as stemming from the closeness of the results for 200 °C and 600 °C. It is noteworthy that these conjectures were also corroborated by the analysis of complementary cumulative distribution functions [50]. However, a detailed comparison of the histograms of statistical distributions would require more experiments and will be presented elsewhere.” The experimental results are mixed with their interpretation. It does not help to distinguish duly the real evidences from their discussion.
Page 7, figure 3. The trend for the curves with the temperature is not duly explained. For sure, the testing at 300, 400 and 700 C give rise to a specific response. However, the reason is far from clear explanation.
Page 9, figure 5. The curves illustrate the trend for type I points. What is about type II?
In addition, temperatures of 300, 400 and 700 C are special point. However, this is not properly explained.
Page 10, line 330. “Let us first recall that due to the combibnation of different approaches to complex signals, dislocation self-organization was detected for all deformation conditions”. What are the real proves of dislocation self-organization?
Page 10, lines 336-339 “The increase itself is conform to the thermally activated motion of dislocations: the higher the temperature, the more deformation events can be activated by the internal stresses generated by a triggering process, e.g., upon a breakthrough of a dislocation pile-up, thus giving rise to a larger variety of correlated sequences of events”. Where are the evidences? The higher the temperature, the more intensive thermal activation. However, the effect of temperature exhibits a different trend.
Page 11, Conclusion “In summary, the first results of the analysis of fine-scale stress fluctuations during tensile deformation of a HEA allow to extend the conclusion on the intrinsically collective nature of the underlying dislocation dynamics. This conclusion was advanced on the basis of investigations of diverse materials using higher-resolution techniques, such as acoustic emission or visualization of the local strain fields. The analysis of stress fluctuations shows that this conjecture covers an even larger range of small scales. It turns out to be a general feature that the macroscopic plastic flow is not a result of averaging over uncorrelated movements of individual dislocations, but over collective dislocation processes taking place on a hierarchical range of mesoscopic scales. Fundamentally, the data obtained testify that the collective effects may manifest qualitatively distinct dynamics in different scale ranges”. This is not a Conclusion but rather concluding remarks.
General remark. The serration at “noise” level must be discussed only in combination with higher scale serration (being mostly exhibited at the temperature of 500 C) as well as temperature effect on the general pattern of stress-strain curve. In the current form, the noise type serration is discussed separately, mostly in terms of dislocation dynamics, that was not studied microscopically.
Round 2
Reviewer 2 Report
The authors have revised the manuscript based on my previous comments. Hence, I am recommending to accept the revised version
Reviewer 5 Report
Thank you for providing a detailed response for every question and remark. The manuscript has been duly revised and improved. It will be decently of interest for the wide audience of the journal on Metals.